# Toward Super-Resolution Image Construction Based on Joint Tensor Decomposition

**Xiaoxu Ren** [1], **Liangfu Lu** [2,*] **and Jocelyn Chanussot** [3]

1    College of Intelligence and Computing, Tianjin University, Tianjin 300350, China; xiaoxuren@tju.edu.cn
2    School of Mathematics, Tianjin University, Tianjin 300350, China
3    LJK, CNRS, Inria, Grenoble INP,  Université Grenoble Alpes, 38000 Grenoble, France;
     jocelyn.chanussot@grenoble-inp.fr
*    Correspondence: liangfulv@tju.edu.cn

**Abstract:** In recent years, fusing hyperspectral images (HSIs) and multispectral images (MSIs) to acquire super-resolution images (SRIs) has been in the spotlight and gained tremendous attention. However, some current methods, such as those based on low rank matrix decomposition, also have a fair share of challenges. These algorithms carry out the matrixing process for the original image tensor, which will lose the structure information of the original image.  In addition, there is no corresponding theory to prove whether the algorithm can guarantee the accurate restoration of the fused image due to the non-uniqueness of matrix decomposition. Moreover, degenerate operators are usually unknown or difficult to estimate in some practical applications. In this paper, an image fusion method based on joint tensor decomposition (JTF) is proposed, which is more effective and more applicable to the circumstance that degenerate operators are unknown or tough to gauge. Specifically, in the proposed JTF method, we consider SRI as a three-dimensional tensor and redefine the fusion problem with the decomposition issue of joint tensors. We then formulate the JTF algorithm, and the experimental results certify the superior performance of the proposed method in comparison to the current popular schemes.

**Keywords:** hyperspectral image; multispectral image; image fusion; joint tensor decomposition

---

## 1. Introduction

With the flourishing of both artificial intelligence (AI) and mathematical theory, image fusion has always been the focus and hotspot in neuroscience, metabonomics, remote sensing, and many other fields [1–3]. Generally, image fusion refers to synthesizing images' data obtained from the diverse image acquisition equipment.  It is aimed at achieving complementary information from different information sources to further acquire clearer, more informative, and higher quality reconstructed images. In 1994, Genderen and Phol proposed a simple and intuitive definition of image fusion [4]: image fusion is merging two or more images into a new image using some algorithms.  Therein, hyperspectral images (HSIs), playing a catalytic role in image fusion, have been widely leveraged in geophysical exploration [5], agricultural remote sensing [6], marine remote sensing [7], environmental monitoring [8], and other fields [9–13] because of their rich spectral information. However, the spatial resolution of HSIs is still relatively low, subjected to the imaging equipment of HSIs and the complex imaging environment, which cannot meet the application requirements of mixing, classification, detection, etc., while this further limits the prospect of HSIs. Therefore, how to improve the resolution of hyperspectral images has become the hot issue in the field of image processing.

Specially, panchromatic fusion and hyperspectral-multispectral fusion are two forms of hyperspectral super-resolution image reconstruction.  Panchromatic fusion refers to fusing multispectral images (or hyperspectral images) and panchromatic images to obtain images with

more spatial information. In [14], the authors classified the pansharpening techniques into component substitution (CS) [15] and multi-resolution analysis (MRA) [16]. Meanwhile, they proposed a hybrid method, combining the better spatial information of CS and the more accurate spectral information of MRA techniques, to improve the spatial resolution while preserving the original spectral information as much as possible.

Concretely, a multispectral image (MSI) generally consists of dozens of bands, and most of them are in the range of the visible region. Given a low-spatial resolution HSI, the operation of spatial resolution enhancement using MSI under the same scene is termed hyperspectral image fusion or super-resolution image (SRI) reconstruction. Generally, the MSI has a higher spatial resolution than the HSI, which is complementary to the HSI. In this paper, we mainly study the method of SRI reconstruction by combining the spectral information of HSI with the spatial information of MSI under the same scene.

Nevertheless, the existing technologies can neither avoid the distortion of image spectral characteristics, nor the complex and time-consuming frequency decomposition and reconstruction. Therefore, Yokoya proposed a simple spectral preservation fusion technique: the smoothing filter based intensity modulation (**SFIM**) [17], which is also called the generalized Laplace pyramid (**GLP**) [18–21] based on the modulation transfer function (**MTF**) used to fuse HSI and MSI by hypersharpening [22]. Then, utilizing the ratio between the high-resolution image and its low-pass filter (with a smoothing filter) image, spatial details could be modulated into co-registered low-resolution MSIs without changing their spectral characteristics and contrast. Compared with Brovey transform [23], **SFIM** is an advanced fusion technology to improve the spatial details of MSIs, and its spectral characteristics are reliably preserved.

Beyond that, Eismann introduced a maximum a posteriori estimation method [24]. It combined the stochastic mixing model of the content under the spectral scene and developed a cost function that could optimize the estimation of the hyperspectral scene related to the observed hyperspectral and auxiliary images. Moreover, this method can generally reconstruct sub-pixel information of several main components in SRI estimation. Furthermore, sparse representation has often been employed to deal with various types of image processing problems, especially in the inverse problem. In 2006, Elad denoised the image with the sparse representation method [25]. This not only achieved the state-of-the-art effect, but introduced the K-SVD dictionary training method [26]. In 2010, the authors of [27] proposed a single-frame super-resolution image reconstruction method based on sparse representation.

In contrast, traditional methods, such as principal component substitution and enhancement of least squares estimation, are primarily limited to the first principal component. For remote sensing images, the spectral characteristics of pixels are denoted as endmembers, including mixed endmembers and pure endmembers. Since each pixel has mixed endmembers, unmixing is a technique for estimating the number of pure endmembers in each pixel, the spectral characteristics, and the abundance of the endmembers [28]. The SRI reconstruction method based on unmixing usually decomposes the HSI and the MSI of the same scene. The endmember matrix of the decomposed HSI and the abundance matrix of the MSI are combined to obtain the reconstructed HSI with high spatial resolution.

In the effort of [29], the authors proposed the method of enhancing the spatial resolution of the HSI in terms of unmixing technology: coupled nonnegative matrix factorization (**CNMF**). They exploited nonnegative matrix factorization to unmix the HSI and MSI sequentially and iteratively obtained the endmember matrix and abundance matrix. Ultimately, the SRI was obtained by combining the two matrices. The nonnegative matrix decomposition usually cannot guarantee the unique solution, although **CNMF** could achieve good reconstruction results. To address this matter, Eliot Wycoff presented a nonnegative sparse enhancement model for SRI reconstruction [30], which testified that the solution was not unique in the **CNMF** method, and it had high computational complexity and a high requirement for CPU operation ability. To further boost the effect of super-resolution reconstruction, the authors of [31] came up with a method to resolve the problem of super-resolution

and hyperspectral unmixing simultaneously. Unlike the measures in [29], they took advantage of the nearest alternating linear minimization (PALM) [32] to update them simultaneously, while the initialization of the endmember matrix applied SISAL [33] for endmember extraction.

Simultaneously, some researchers studied the fusion of the MSI and HSI based on the tensor [34–40], mainly considering the natural tensor structure of spectral images, so as to reduce the information lost in matricization and increase the performance. Generally, multi-channel images and other data have their own natural tensor structure. In addition, since the tensor has good expressive ability and computational characteristics, it is very meaningful to study the tensor analysis of images. Moreover, tensor decomposition can preserve the structural characteristics of the original image data. For HSIs, tensor decomposition makes full use of spatial and spectral redundancy between images and compresses and extracts relevant feature information with high quality. Based on HSIs, a nonnegative tensor canonical polyadic decomposition (CP) algorithm was raised that was applied to dispose of the blind source separation [41]. Shashua utilized CP decomposition for image compression and classification [42], while Bauckhage introduced discriminant analysis to high-order data such as color images for classification [43]. Xiao Fu proposed a coupled tensor decomposition framework [44], which could guarantee the identifiability of SRIs under mild and realistic conditions. Meanwhile, Shutao Li and Renwei Dian put forward a coupled sparse tensor factorization (**CSTF**) [45]; they regarded the SRI as a three-dimensional tensor and redefined the fusion problem as the core tensor and dictionary estimation of three modes. The high-spatial spectral correlation in the SRI was modeled by a regularizer, which could promote the generation of sparse core tensors. However, **CSTF** is an optimization model based on tensor Tucker decomposition, which is not unique. Moreover, most existing methods assume that known (or easily estimated) degenerate operators are applied to SRI to form the corresponding HSI and MSI, which is practically absent. In this paper, we deal with the super-resolution problem under the condition that the degenerate operators are seldom known and contain noise. A joint tensor decomposition model is proposed by taking advantage of the multi-dimensional tensor structure of the HSI and MSI.

The main content of this paper is to utilize the joint tensor decomposition (**JTF**) algorithm for the fusion of HSI and MSI, so as to explore the problem of SRI reconstruction. The contributions are listed as follows:

- In the proposed method, the high-spatial resolution HSI is regarded as a three-dimensional tensor, while the fusion issue is redefined as the joint estimation of the coupling factor matrix, which is also expressed as the joint tensor decomposition problem for the hyperspectral image tensor, multispectral image tensor, and noise regularization term.
- In order to observe the reconstruction effect of this method, the performance of this algorithm is compared with the five algorithms. Experiments show that the **JTF** method provides clearer spatial details than the real SRI, while the running time of **JTF** is acceptable compared with the excellent performance.
- Besides, we conduct experiments under the incorrect Gaussian kernel ($3 \times 3$, $5 \times 5$, $7 \times 7$), correct Gaussian kernel ($9 \times 9$), and different noises, while showing the fusion effect of the six test methods with the Pavia University data captured by the ROSISsensor as well. The results reveal that the **JTF** method performs best in comparison with the other methods in terms of reconstruction accuracy regardless of whether the Gaussian kernel is correct and the level of added noise. This indicates that the **JTF** algorithm is more suitable for degradation operators that are unknown or contain noise.

The outline of this paper is organized as follows. In Section 1, we mainly introduce some basic notations and definitions for tensors. In Section 2, we give a basic overview of tensors. The proposed coupled image fusion algorithms are introduced in Section 3. In Section 4, experimental results on the algorithms are presented. Conclusions and future research directions are given in Section 6.

## 2. Preliminaries on Tensors

### 2.1. Definition and Notations

In this section, we first briefly introduce some necessary notions and preliminaries. The general tensor is denoted as $\mathcal{X}$, while the element $(i, j, k)$ of a third-order tensor $\mathcal{X}$ is signed by $x_{ijk}$. The matrix is denoted as $\mathbf{X}$, and the scalar (or the vector) is represented by $x$. A fiber is defined by fixing every index but one. Third-order tensors have column, row, and tube fibers, denoted by $x_{:jk}$, $x_{i:k}$, and $x_{ij:}$, respectively; see Figure 1. Fibers are always assumed to be column vectors. The mode-$n$ matricization of a tensor $\mathcal{X} \in \mathbb{R}^{I_1 \times I_2 \times \cdots \times I_N}$ is signed by $\mathbf{X}_{(n)}$, which arranges the mode-n fibers to be the columns of the matrix and can reduce the dimension of the tensor.

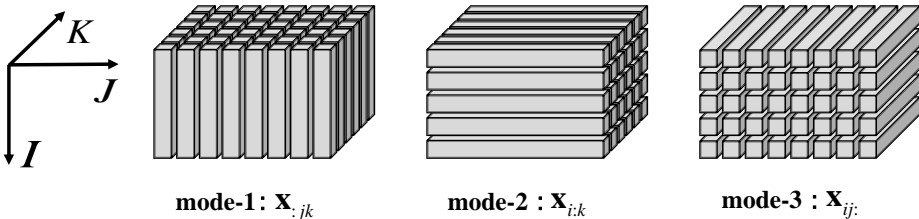

**mode-1 : $\mathbf{X}_{:jk}$**　　　　**mode-2 : $\mathbf{X}_{i:k}$**　　　　**mode-3 : $\mathbf{X}_{ij:}$**

**Figure 1.** Fibers of a third-order tensor .

**Definition 1.** *The Kronecker product of matrices $\mathbf{A} \in \mathbb{R}^{I \times J}$ and $\mathbf{B} \in \mathbb{R}^{K \times L}$ is defined by Equation (1), which is denoted as $\mathbf{A} \otimes \mathbf{B}$, and the calculation result is a matrix of size $IK \times JL$, i.e.,*

$$\mathbf{A} \otimes \mathbf{B} = \begin{bmatrix} a_{11}\mathbf{B} & a_{12}\mathbf{B} & \cdots & a_{1J}\mathbf{B} \\ a_{21}\mathbf{B} & a_{22}\mathbf{B} & \cdots & a_{2J}\mathbf{B} \\ \vdots & \vdots & \ddots & \vdots \\ a_{I1}\mathbf{B} & a_{I2}\mathbf{B} & \cdots & a_{IJ}\mathbf{B} \end{bmatrix} \tag{1}$$

$$= [a_1 \otimes b_1 \quad a_1 \otimes b_2 \quad a_1 \otimes b_3 \cdots a_J \otimes b_{L-1} \quad a_J \otimes b_L].$$

Then, we have a new matrix-matrix production termed as the Khatri–Rao product.

**Definition 2.** *Let $\mathbf{A} \in \mathbb{R}^{I \times K}$ and $\mathbf{B} \in \mathbb{R}^{J \times K}$. Then, the Khatri–Rao product is a matrix of size $IJ \times K$ defined as:*

$$\mathbf{A} \odot \mathbf{B} = [a_1 \otimes b_1 \quad a_2 \otimes b_2 \quad \cdots \quad a_K \otimes b_K], \tag{2}$$

*where $\otimes$ is the Kronecker product.*

Next, we discuss some properties of the Khatri–Rao product, which will be useful in our later discussion.

$$\mathbf{A} \odot \mathbf{B} \odot \mathbf{C} = (\mathbf{A} \odot \mathbf{B}) \odot \mathbf{C} = \mathbf{A} \odot (\mathbf{B} \odot \mathbf{C}),$$
$$(\mathbf{A} \odot \mathbf{B})^{\mathrm{T}}(\mathbf{A} \odot \mathbf{B}) = \mathbf{A}^{\mathrm{T}}\mathbf{A} * \mathbf{B}^{\mathrm{T}}\mathbf{B}, \tag{3}$$
$$(\mathbf{A} \odot \mathbf{B})^{\dagger} = ((\mathbf{A}^{\mathrm{T}}\mathbf{A}) * (\mathbf{B}^{\mathrm{T}}\mathbf{B}))^{\dagger}(\mathbf{A} \odot \mathbf{B})^{\mathrm{T}}.$$

**Definition 3.** *The n-mode (matrix) product of a tensor $\mathcal{X} \in \mathbb{R}^{I_1 \times I_2 \times \cdots \times I_N}$ with a matrix $\mathbf{M} \in \mathbb{R}^{J \times I_n}$ is represented as:*

$$(\mathcal{X} \times_n \mathbf{M})_{i_1 \cdots i_{n-1} j i_{n+1} \cdots i_N} = \sum_{i_n=1}^{i_N} x_{i_1 i_2 \cdots i_N} m_{j i_n}, \tag{4}$$

*which can be denoted by $\mathcal{X} \times_n \mathbf{M}$ and is a tensor with a size of $I_1 \times \cdots \times I_{n-1} \times J \times I_{n+1} \times \cdots \times I_N$.*

**Definition 4.** *The Frobenius norm of a tensor* $\mathcal{X} \in \mathbb{R}^{I_1 \times I_2 \times \cdots \times I_N}$ *is represented as:*

$$\| \mathcal{X} \| = \sqrt{\sum_{i_1=1}^{I_1} \sum_{i_2=1}^{I_2} \cdots \sum_{i_N=1}^{I_N} x_{i_1,i_2 \cdots iN}^2}. \tag{5}$$

*2.2. Tensor Decomposition*

The general tensor decomposition models involve CP decomposition and Tucker decomposition. Specifically, the CP decomposition is a special case of the Tucker decomposition. Due to the special structure of tensors, these tensor decomposition methods are leveraged in hyperspectral image processing. For a tensor $\mathcal{X} \in \mathbb{R}^{I_1 \times I_2 \times \cdots \times I_N}$, the CP decomposition could be expressed as:

$$\mathcal{X} \approx \sum_{r=1}^{R} \lambda_r a_r^{(1)} \circ a_r^{(2)} \circ \cdots \circ a_r^{(N)} = [\![\lambda; \mathbf{A}^{(1)}, \mathbf{A}^{(2)}, \cdots, \mathbf{A}^{(N)}]\!], \tag{6}$$

where "$\circ$" is the outer product of the vectors, $R$ is a positive integer, and $\mathbf{A}^{(n)}$ is the factor matrix. For $n = 1, 2, \cdots, N$, $\lambda \in \mathbb{R}^R$, $a_r^{(n)} \in \mathbb{R}^{I_n}$, $\mathbf{A}^{(n)} \in \mathbb{R}^{I_n \times R}$, the factor matrix is a combination of the rank one vector $a_r^{(n)}$ and denoted as:

$$\mathbf{A}^{(n)} = [a_1^{(n)}, a_2^{(n)}, \cdots, a_R^{(n)}], \tag{7}$$

Let the three-order tensor $\mathcal{X} \in \mathbb{R}^{I \times J \times K}$ be a hyperspectral image, the CP decomposition could be formulated as:

$$\mathcal{X} \approx \sum_{r=1}^{R} \lambda_r a_r \circ b_r \circ c_r = [\![\lambda; \mathbf{A}, \mathbf{B}, \mathbf{C}]\!], \tag{8}$$

where $I$, $J$, and $K$ are the numbers of the row, column, and spectral dimensions, respectively, while $r = 1, 2, \cdots, R$, $\lambda \in \mathbb{R}^R$, $a_r \in R^I$, $b_r \in R^J$, $c_r \in R^K$.

Each column of the above factor matrices $\mathbf{A}$, $\mathbf{B}$, and $\mathbf{C}$ is normalized, and $\lambda_r$ is the weight. If there is no requirement to standardize the factor matrix, the CP decomposition can also be reformulated as:

$$\mathcal{X} \approx (\mathbf{A}', \mathbf{B}', \mathbf{C}'). \tag{9}$$

where $\mathbf{A}', \mathbf{B}', \mathbf{C}'$ mean the general factor matrices, which are constructed by assigning the weight to the factor matrices $\mathbf{A}, \mathbf{B}, \mathbf{C}$.

The schematic diagram of CP decomposition is shown in Figure 2. If $R$ denotes the minimum number of outer products needed to express $\mathcal{X}$, then the tensor rank is $R$, i.e., $rank(\mathcal{X}) = R$, and the decomposition is known as rank decomposition, which is a particular case of CP decomposition. At present, there is no specific method to directly solve the rank of any given tensor, which has been proven to be an NP-hard problem. Through the factor matrix, the CP decomposition of a third-order tensor can be written in expansion form.

$$\begin{aligned} \mathbf{X}_{(1)} &= \mathbf{A}'(\mathbf{C}' \odot \mathbf{B}')^{\mathrm{T}}, \\ \mathbf{X}_{(2)} &= \mathbf{B}'(\mathbf{C}' \odot \mathbf{A}')^{\mathrm{T}}, \\ \mathbf{X}_{(3)} &= \mathbf{C}'(\mathbf{B}' \odot \mathbf{A}')^{\mathrm{T}}. \end{aligned} \tag{10}$$

A third-order tensor can be denoted as follows by applying mode-n products:

$$\mathcal{X}' = \mathcal{X} \times_1 \mathbf{D_1} \times_2 \mathbf{D_2} \times_3 \mathbf{D_3}, \tag{11}$$

The above formula can be expressed in the form of factor matrices:

$$\mathcal{X}' \approx (\mathbf{D_1}\mathbf{A}', \mathbf{D_2}\mathbf{B}', \mathbf{D_3}\mathbf{C}'). \tag{12}$$

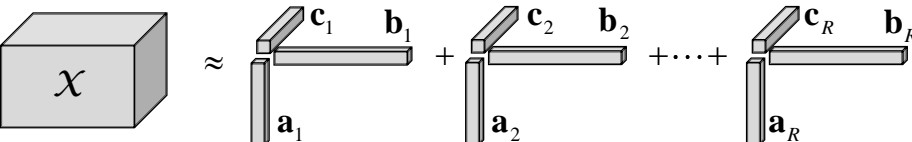

**Figure 2.** Canonical polyadic (CP) decomposition of third-order tensors .

**Theorem 1.** *Correspondingly, we consider how many rank-one tensors (components) of the decomposition of the CP model are added to minimize the error. The usual practice is to start with $R = 1$ until you encounter a "good" result. Of course, if you have a strong application background and prior information, you can also specify it in advance. For a given number of components, there is still no universal solution for CP decomposition. Specifically, the alternating least squares (ALS) algorithm is a more popular method in the case that the number of components is pre-given [46]. For the CP decomposition of tensors, even if R is much larger than $max\{i, j, k\}$, the CP decomposition model is essentially unique. The lower order decomposition of matrices and Tucker decomposition of tensors are generally not unique, which is a significant difference between them. The most famous result about the uniqueness of tensor decomposition is due to Kruskal [47]. One result of the Kruskal criteria is the following statement, which applies to general tensors, which provides the uniqueness proof of the CP decomposition model.*

*Suppose $\mathcal{X} = [\![\mathbf{A}, \mathbf{B}, \mathbf{C}]\!]$ and tensor $\mathcal{X} \in \mathbb{R}^{I \times J \times K}$ of rank R has a unique decomposition if:*

$$R \leq \frac{1}{2}[min(I, R) + min(J, R) + min(K, R) - 2].$$

*where $\mathbf{A} \in \mathbb{R}^{I \times R}$, $\mathbf{B} \in \mathbb{R}^{J \times R}$, and $\mathbf{C} \in \mathbb{R}^{K \times R}$.*

The uniqueness condition of the tensor CP decomposition model is relatively relaxed compared with that of the matrix decomposition model. Since the rank of matrix decomposition must be lower than the dimension of the matrix and needs nonnegative, sparse, and geometric conditions, certainly, we can also judge whether the CP decomposition model is unique according to the rank of the given tensor.

## 3. Problem Formulation

The purpose of HSI and MSI fusion is to estimate the unobservable SRI ($\mathcal{S} \in \mathbb{R}^{I \times J \times K}$) from the observable low-spatial resolution HSI ($\mathcal{H} \in \mathbb{R}^{i \times j \times K}$) and the high-spatial resolution MSI ($\mathcal{M} \in \mathbb{R}^{I \times J \times k}$), where $I(i)$ and $J(j)$ denote the spatial dimensions and $K(k)$ denotes the number of spectral bands. The tensor $\mathcal{H}$ is spatially downsampled with respect to (w.r.t.) $\mathcal{S}$, that is $I > i$ and $J > j$, while the tensor $\mathcal{M}$ is spectrally downsampled w.r.t. $\mathcal{S}$, that is $K > k$. We assume that the two observed data are obtained under the same atmospheric and illumination conditions and are geometrically combined with radiation correction.

### 3.1. Image Fusion Based on Matrix Decomposition

The fusion method based on matrix factorization assumes that each spectral vector of the target SRI can be written as a small number of linear combinations of different spectral characteristics [48], which can be represented as:

$$\mathbf{S}_{(3)} = \mathbf{WH}, \tag{13}$$

where $\mathbf{S}_{(3)} \in \mathbb{R}^{IJ \times K}$ is the three-mode unfolding matrix of the tensor $\mathcal{S}$. Matrices $\mathbf{W} \in \mathbb{R}^{IJ \times R}$ and $\mathbf{H} \in \mathbb{R}^{R \times K}$ represent the spectral basis and the corresponding coefficient matrix, respectively, where $R \ll min\{IJ, K\}$.

The spatial domain of low-spatial resolution hyperspectral data is degraded from the spatial

domain of multispectral data. On the other hand, multispectral data are a form of spectral degradation of high-spatial resolution hyperspectral data. Therefore, $\mathcal{H}$ and $\mathcal{M}$ are modeled as:

$$\mathbf{H}_{(3)} = \mathbf{W}\mathbf{H}_h, \mathbf{M}_{(3)} = \mathbf{W}_m\mathbf{H}, \tag{14}$$

where $\mathbf{H}_h = \mathbf{H}\mathbf{D_H}$, $\mathbf{W}_m = \mathbf{D_M}\mathbf{W}$, $\mathbf{H}_{(3)} \in \mathbb{R}^{ij \times K}$, and $\mathbf{M}_{(3)} \in \mathbb{R}^{IJ \times k}$ are the three-mode unfolding matrices of the HSI (tensor $\mathcal{H}$) and MSI (tensor $\mathcal{M}$), respectively. $\mathbf{D_H} \in \mathbb{R}^{IJ \times ij}$ is a matrix modeling the point spread function (PSF) and the spatial subsampling process in the hyperspectral sensor. $\mathbf{D_M} \in \mathbb{R}^{K \times k}$ is a matrix modeling spectral downsampling in the multispectral sensor, whose rows contain the spectral response of the multispectral sensor. Therefore, the matricized HSI and MSI are modeled as:

$$\mathbf{H}_{(3)} = \mathbf{W}\mathbf{H}\mathbf{D_H}, \mathbf{M}_{(3)} = \mathbf{D_M}\mathbf{W}\mathbf{H}, \tag{15}$$

In the matrix decomposition based fusion approaches, if the spectral basis $\mathbf{D_H}$ and coefficient matrix $\mathbf{D_M}$ can be estimated by jointly factoring from $\mathbf{H}_{(3)}$ and $\mathbf{M}_{(3)}$, the SRI can be restored according to Equation (13), which is the main idea based on matrix decomposition.

### 3.2. Image Fusion Based on Tensor Decomposition

Matrix based methods usually assume that degradation operators are known or easily estimated, but in practice, it is difficult to determine. By comparing the spectral properties of hyperspectral and multispectral sensors, the degradation operator $\mathbf{D_M}$ can be modeled and estimated relatively easily. However, the spatial operator becomes a bit difficult. A common model assumption from SRI to HSI conversion is a combination of the blurring by a Gaussian kernel and a downsampling process. Of course, this is a rough approximation and may be far from accurate. Even if this assumption is approximately correct, there are still many uncertainties.

In order to solve the non-uniqueness of matrix decomposition and under the condition of little knowledge of degenerate operators and noise, we propose a method based on joint tensor decomposition to fuse the HSI and MSI in this section. Tensor based models have many advantages. For example, it is a very efficient strategy to abstract image data into tensor representation and then input them in the image fusion model. For the output data, we can choose the desired format to save them conveniently. Formally, we represent the SRI as the following equation via CP decomposition:

$$\mathcal{S} = [\![\mathbf{A}, \mathbf{B}, \mathbf{C}]\!] \tag{16}$$

where $\mathcal{S} \in \mathbb{R}^{I \times J \times K}$, $\mathbf{A} \in \mathbb{R}^{I \times R}$, $\mathbf{B} \in \mathbb{R}^{J \times R}$, $\mathbf{C} \in \mathbb{R}^{K \times R}$, and $R$ is the the number of components.

The HSI is the spatial downsampling version of the SRI. Assuming that the point spread function (PSF) of the hyperspectral sensor is separable from the downsampling matrix of the wide mode and the high mode, we can have:

$$\mathcal{H} = \mathcal{S} \times_1 \mathbf{D}_1 \times_2 \mathbf{D}_2 \tag{17}$$

where $\mathbf{D}_1 \in \mathbb{R}^{I \times i}$, $\mathbf{D}_2 \in \mathbb{R}^{J \times j}$ are the spatial degradation along the width and height modes, respectively. For subsampling, the separability hypothesis implies that the function of spatial subsampling matrix $\mathbf{D_H}$ is decoupled from the two spatial patterns of $\mathcal{S}$, and thus, the degenerate operator $\mathbf{D_H} = \mathbf{D}_2 \otimes \mathbf{D}_1$ in the matricized form. Under the separability assumption, the HSI ($\mathcal{H}$) can be represented as:

$$\mathcal{H} = [\![\mathbf{A}', \mathbf{B}', \mathbf{C}]\!] \tag{18}$$

where $\mathbf{A}' = \mathbf{D}_1\mathbf{A} \in \mathbb{R}^{I \times R}$, $\mathbf{B}' = \mathbf{D}_2\mathbf{B} \in \mathbb{R}^{J \times R}$, and $\mathbf{C} \in \mathbb{R}^{k \times R}$. In this paper, we assume that spectral response $\mathbf{D_M}$ has noise, i.e., rough sampling in the process of conversion from the SRI to the MSI. Formally, we represent it as:

$$\mathbf{D}'_{\mathbf{M}} = \mathbf{D_M} + \Gamma \tag{19}$$

where $\Gamma$ is Gaussian random noise. Analogously, the MSI ($\mathcal{M}$) can be represented as:

$$\mathcal{M} = \mathcal{S} \times_3 \mathbf{D}_{\mathbf{M}}' \tag{20}$$

where $\mathbf{D}_{\mathbf{M}}' \in \mathbb{R}^{K \times k}$ is the downsampling matrix of the spectral mode. We substitute Formula (16) into (20) to obtain:

$$\mathcal{M} = [\![\mathbf{A}, \mathbf{B}, \mathbf{C}']\!] \tag{21}$$

where $\mathbf{A} \in \mathbb{R}^{i \times R}$, $\mathbf{B} \in \mathbb{R}^{j \times R}$, and $\mathbf{C}' = \mathbf{D}_{\mathbf{M}}'\mathbf{C} \in \mathbb{R}^{K \times R}$. In order to reconstruct the SRI, we need to estimate the factor matrices $\mathbf{A}, \mathbf{B}, \mathbf{C}$.

## 4. The Joint Tensor Decomposition Method

*The Joint Tensor Decomposition Method*

In this section, we consider that when $\mathbf{D}_{\mathbf{M}}$ contains noise and the spatial degradation operator $\mathbf{D}_{\mathbf{H}} = \mathbf{D}_{\mathbf{2}} \otimes \mathbf{D}_{\mathbf{1}}$ is completely unknown, even though this type of operation is called a combination of blurring and downsampling, in practice, hyperparameters such as the blurring kernel type, kernel size, and downsampling offset are barely known. Therefore, the joint tensor decomposition model can be generalized to the following model:

$$\min_{\mathbf{A},\mathbf{B},\mathbf{C}} \| \mathcal{H} - (\mathbf{A}', \mathbf{B}', \mathbf{C}) \|_{\mathrm{F}}^2 + \| \mathcal{M} - (\mathbf{A}, \mathbf{B}, \mathbf{C}') \|_{\mathrm{F}}^2 + \beta \| \mathbf{C}' - \mathbf{D}_{\mathbf{M}}'\mathbf{C} \|_{\mathrm{F}}^2, \tag{22}$$

We use the following optimization models to obtain factor matrices $\mathbf{A}, \mathbf{B}$ and $\mathbf{C}$, where $\beta$ is the regularization parameter. The above optimization problem is non-convex, and the solutions of the factor matrices $\mathbf{A}, \mathbf{B}$, and $\mathbf{C}$ are not unique. However, the objective function in (22) is convex for each variable block, remaining unchanged with other variables. Therefore, we choose the proximal alternate optimization (PAO) scheme to solve the above optimization problem, which guarantees that the optimization problem converges to the critical point under certain conditions. Then, each step of the iterative update of the factor matrix is reduced to solving an easy-to-handle Sylvester equation by matricization of tensor HSI and MSI. Specifically, the $\mathbf{A}, \mathbf{B}$, and $\mathbf{C}$ iterations are updated as follows:

- Optimization with respect to $\mathbf{C}$:

When $\mathbf{A}, \mathbf{B}, \mathbf{A}', \mathbf{B}'$, and $\mathbf{C}'$ are fixed, the optimization w.r.t. $\mathbf{C}$ in (22) can be written as:

$$\min_{\mathbf{C}} \| \mathcal{H} - (\mathbf{A}', \mathbf{B}', \mathbf{C}) \|_{\mathrm{F}}^2 + \| \mathcal{M} - (\mathbf{A}, \mathbf{B}, \mathbf{C}') \|_{\mathrm{F}}^2 + \beta \| \mathbf{C}' - \mathbf{D}_{\mathbf{M}}'\mathbf{C} \|_{\mathrm{F}}^2,$$

The above optimization problem can be transformed into the following one by using the properties of n-mode matrix unfolding.

$$\min_{\mathbf{C}} \| \| \mathbf{H}_{(3)} - \mathbf{C}(\mathbf{B}' \odot \mathbf{A}')^{\mathrm{T}}) \|_{\mathrm{F}}^2 + \beta \| \mathbf{C}' - \mathbf{D}_{\mathbf{M}}'\mathbf{C} \|_{\mathrm{F}}^2 \tag{23}$$

where $\mathbf{H}_{(3)}$ is the three-mode unfolding matrix of tensors $\mathcal{H}$. The optimization problem (23) is quadratic, and its unique solution is equal to the calculation of the general Sylvester matrix equation.

$$\beta \mathbf{D}_{\mathbf{M}}'^{\mathrm{T}}\mathbf{D}_{\mathbf{M}}'\mathbf{C} + \mathbf{C}\mathbf{E} - \beta \mathbf{D}_{\mathbf{M}}'^{\mathrm{T}}\mathbf{C}' = \mathbf{H}_{(3)}\mathbf{E} \tag{24}$$

where $\mathbf{E} = (\mathbf{B}'^{\mathrm{T}}\mathbf{B}') * (\mathbf{A}'^{\mathrm{T}}\mathbf{A}')$.

We use the Sylvester function in the MATLAB toolbox to solve the above equation.

- Optimization with respect to $\mathbf{A}'$:

When $\mathbf{A}$, $\mathbf{B}$, $\mathbf{C}$, $\mathbf{B}'$, and $\mathbf{C}'$ are fixed, the optimization w.r.t. $\mathbf{A}'$ in (22) can be written as:

$$\min_{\mathbf{A}'} \| \mathcal{H} - (\mathbf{A}', \mathbf{B}', \mathbf{C}) \|_{\mathrm{F}}^2, \tag{25}$$

The above optimization problem can be transformed into the following one by using the properties of n-mode matrix unfolding.

$$\min_{\mathbf{A}'} \| \mathbf{H}_{(1)} - \mathbf{A}'(\mathbf{C} \odot \mathbf{B}')^{\mathrm{T}}) \|_{\mathrm{F}}^2, \tag{26}$$

where $\mathbf{H}_{(1)}$ is the one-mode unfolding matrix of tensors $\mathcal{H}$. The optimization problem (26) is convex, and the optimal solution is then given by:

$$\mathbf{A}' = \mathbf{H}_{(1)}[(\mathbf{C} \odot \mathbf{B}')^{\mathrm{T}}]^{\dagger}, \tag{27}$$

According to the property of the Khatri–Rao product pseudo-inverse, we can rewrite the solution as:

$$\mathbf{A}' = \mathbf{H}_{(1)}(\mathbf{C} \odot \mathbf{B}')(\mathbf{C}^{\mathrm{T}}\mathbf{C} * \mathbf{B}'^{\mathrm{T}}\mathbf{B}')^{\dagger}, \tag{28}$$

The advantage of solving the above equation is that we only need to compute the pseudo-inverse matrix of the $R \times R$ matrix, but not the $jK \times R$ matrix. The solving process of factor matrix $\mathbf{B}'$ is similar to that of $\mathbf{A}'$, and we can rewrite the solution as:

$$\mathbf{B}' = \mathbf{H}_{(2)}(\mathbf{C} \odot \mathbf{A}')(\mathbf{C}^{\mathrm{T}}\mathbf{C} * \mathbf{A}'^{\mathrm{T}}\mathbf{A}')^{\dagger}. \tag{29}$$

- Optimization with respect to $\mathbf{C}'$:

When $\mathbf{A}$, $\mathbf{B}$, $\mathbf{C}$, $\mathbf{A}'$, and $\mathbf{B}'$ are fixed, the optimization w.r.t. $\mathbf{C}'$ in (22) can be written as the following one by using the properties of n-mode matrix unfolding.

$$\min_{\mathbf{C}'} \| \mathbf{M}_{(3)} - \mathbf{C}'(\mathbf{B} \odot \mathbf{A})^{\mathrm{T}}) \|_{\mathrm{F}}^2 + \beta \| \mathbf{C}' - \mathbf{D}'_{\mathbf{M}}\mathbf{C} \|_{\mathrm{F}}^2 \tag{30}$$

where $\mathbf{M}_{(3)}$ is the three-mode unfolding matrix of tensors $\mathcal{M}$. The optimization problem (30) is quadratic, and its unique solution is equal to the calculation of the general Sylvester matrix equation.

$$\beta \mathbf{I}^{\mathrm{T}}\mathbf{I}\mathbf{C}' + \mathbf{C}'\mathbf{F} - \beta \mathbf{D}'_{\mathbf{M}}\mathbf{C} = \mathbf{H}_{(3)}\mathbf{F} \tag{31}$$

where $\mathbf{F} = (\mathbf{B}^{\mathrm{T}}\mathbf{B}) * (\mathbf{A}^{\mathrm{T}}\mathbf{A})$, and $\mathbf{I}$ is the unit matrix of $4 \times 4$.

We use the Sylvester function in the MATLAB toolbox to solve the above equation.

- Optimization with respect to $\mathbf{A}$:

When $\mathbf{B}$, $\mathbf{C}$, $\mathbf{A}'$, $\mathbf{B}'$, and $\mathbf{C}'$ are fixed, the optimization w.r.t. $\mathbf{A}$ in (22) can be written as:

$$\min_{\mathbf{A}} \| \mathcal{M} - (\mathbf{A}, \mathbf{B}, \mathbf{C}') \|_{\mathrm{F}}^2, \tag{32}$$

The above optimization problem can be transformed into the following one by using the properties of n-mode matrix unfolding.

$$\min_{\mathbf{A}} \| \mathbf{M}_{(1)} - \mathbf{A}(\mathbf{C}' \odot \mathbf{B})^{\mathrm{T}} \|_{\mathrm{F}}^2, \tag{33}$$

where $\mathbf{M}_{(1)}$ is the one-mode unfolding matrix of tensors $\mathcal{M}$. The optimization problem (33) is convex, and the optimal solution is then given by:

$$\mathbf{A} = \mathbf{M}_{(1)}[(\mathbf{C}' \odot \mathbf{B})^{\mathrm{T}}]^{\dagger}, \tag{34}$$

According to the property of the Khatri–Rao product pseudo-inverse, we can rewrite the solution as:

$$\mathbf{A} = \mathbf{M}_{(1)}(\mathbf{C}' \odot \mathbf{B})(\mathbf{C}'^{\mathrm{T}}\mathbf{C} * \mathbf{B}'^{\mathrm{T}}\mathbf{B})^{\dagger}, \tag{35}$$

Similarly, we only need to compute the pseudo-inverse matrix of the $R \times R$ matrix, but not the $Jk \times R$ matrix. The solving process of factor matrix $\mathbf{B}$ is similar to that of $\mathbf{A}$, and we can rewrite the solution as:

$$\mathbf{B} = \mathbf{M}_{(2)}(\mathbf{C}' \odot \mathbf{A})(\mathbf{C}'^{\mathrm{T}}\mathbf{C}' * \mathbf{A}^{\mathrm{T}}\mathbf{A})^{\dagger}, \tag{36}$$

For each iteration update, we discuss it in detail. The specific algorithm is shown in Algorithm 1. After obtaining the estimated values of $\mathbf{A}$, $\mathbf{B}$, and $\mathbf{C}$, the super-resolution tensor reconstruction is obtained from the following formula:

$$\mathcal{S} \approx (\mathbf{A}, \mathbf{B}, \mathbf{C}). \tag{37}$$

The detailed steps of the proposed method are given in Algorithm 1.

---

**Algorithm 1:** Algorithm for coupled images.

**Initialization:** $\beta, R, \mathbf{A_0}, \mathbf{B_0}, \mathbf{C_0}, \mathbf{A}'_0, \mathbf{B}'_0, \mathbf{C}'_0$
  Apply blind STEREO Algorithm [44] with random
  initializations to obtain $\mathbf{A}, \mathbf{B}, \mathbf{C}, \mathbf{A}', \mathbf{B}', \mathbf{C}'$
**While** not converged, do
  $\mathbf{C} \leftarrow \arg\min_{\mathbf{C}} \| \mathbf{H}_{(3)} - \mathbf{C}(\mathbf{B}' \odot \mathbf{A}')^{\mathrm{T}}) \|_{\mathrm{F}}^2 + \beta \| \mathbf{C}' - \mathbf{D}'_{\mathbf{M}}\mathbf{C} \|_{\mathrm{F}}^2$
  $\mathbf{A}' \leftarrow \arg\min_{\mathbf{A}'} \| \mathbf{H}_{(1)} - \mathbf{A}'(\mathbf{C} \odot \mathbf{B}')^{\mathrm{T}}) \|_{\mathrm{F}}^2,$
  $\mathbf{B}' \leftarrow \arg\min_{\mathbf{B}'} \| \mathbf{H}_{(2)} - \mathbf{B}'(\mathbf{C} \odot \mathbf{A}')^{\mathrm{T}}) \|_{\mathrm{F}}^2,$
  $\mathbf{C}' \leftarrow \arg\min_{\mathbf{C}'} \| \mathbf{M}_{(3)} - \mathbf{C}'(\mathbf{B} \odot \mathbf{A})^{\mathrm{T}}) \|_{\mathrm{F}}^2 + \beta \| \mathbf{C}' - \mathbf{D}'_{\mathbf{M}}\mathbf{C} \|_{\mathrm{F}}^2,$
  $\mathbf{A} \leftarrow \arg\min_{\mathbf{A}} \| \mathbf{M}_{(1)} - \mathbf{A}(\mathbf{C}' \odot \mathbf{B})^{\mathrm{T}}) \|_{\mathrm{F}}^2,$
  $\mathbf{B} \leftarrow \arg\min_{\mathbf{B}} \| \mathbf{M}_{(2)} - \mathbf{B}(\mathbf{C}' \odot \mathbf{A})^{\mathrm{T}}) \|_{\mathrm{F}}^2.$
**end while**

---

## 5. Experiments And Results

### 5.1. Experimental Data

To obtain an MSI from an SRI, we used the spectral specifications of the multispectral sensor, which were taken from the QuickBird sensor in our experiments [49]. The QuickBird sensor produces four-band MSI in the following spectral bands: blue (430–545 nm), green (466–620 nm), red (590–710 nm), and near-infrared (715–918 nm). Then, the spectral response matrix $\mathbf{D_M}$ is formed by comparing the SRI obtained in the experiment from 400 to 2500 nm with the multi-spectral sensor band, and the MSI image is obtained by assuming that there is random Gaussian noise in $\mathbf{D_M}$. More precisely, $\mathbf{D_M}$ is a selective averaging matrix that acts on the common wavelength of the SRI and MSI, and the experimental data came from [44]. The data selected in this paper were taken from Pavia University in Italy and were captured by the ROSIS sensor. The SRI, HSI, and MSI have sizes of $608 \times 336 \times 103$, $152 \times 84 \times 103$, and $608 \times 336 \times 4$, respectively. Specifically, the MSI is generated by QuickBird simulation, while the HSI is generated by SRI by $9 \times 9$ Gaussian blur and downsampling, and the MSI is generated for the Pavia University image according to the QuickBird specification. The degradation process from the SRI to the HSI is a combination of spatial blurring of the $9 \times 9$ Gaussian kernel and the $D = 4$ factor along two spatial directions to model the blurred image.

On the 3.6 GHz kernel and 8 GB RAM Windows server, the simulation is carried out by MATLAB. According to the algorithm **JTF**, factors $\mathbf{A}$, $\mathbf{B}$, and $\mathbf{C}$ are obtained through the joint tensor decomposition of the MSI and HSI, which mainly solves the least squares problem and preliminarily estimates

the potential factors, where the CP decomposition is computed by TensorLab [50]. The maximum number of iterations for tensor decomposition was set to 25 in the initialization, while the number of iteration updates for factor matrix requires continuing numerical simulation. In this paper, we fixed $\beta = 1$. We mainly refer to [44]; this paper proved that the performance of super-resolution tensor reconstruction is best when beta is equal to one. The proposed model adds the noise term in the objective function based on [44]. Therefore, we selected a similar parameter.

To further demonstrate the performance of our proposed algorithm, this method is compared with the following five HSI-MSI fusion methods: **Blind Stereo** [44], **CNMF** (coupled nonnegative matrix factorization) [29], **SFIM** (smoothing filter based intensity modulation) [51], **MTF-GLP** (modulation transfer function based generalized Laplacian pyramid) [19], and **MAPSMM** (maximum a posterior estimation with a stochastic mixing model) [24].

*5.2. Evaluation Criterion*

In order to evaluate the quality of reconstructed high-spatial resolution HSIs, we introduce several intuitive evaluation indicators. The first index is the reconstruction signal-to-noise ratio (R-SNR) criterion defined as:

$$\text{R} - \text{SNR} = 10\log_{10}\left(\frac{\sum_{k=1}^{K}\|\mathbf{S_k}\|_F^2}{\sum_{k=1}^{K}\|\mathbf{S'_k} - \mathbf{S_k}\|_F^2}\right). \tag{38}$$

where $\mathbf{S'_k}$ and $\mathbf{S_k}$ are the frontal slices of reconstructed SRI and ground truth SRI. The higher the R-SNR is, the better the reconstruction quality.

The second index is the root mean squared error (RMSE), i.e.,

$$\text{RMSE} = \sqrt{\frac{\|\mathcal{S}' - \mathcal{S}\|_F^2}{\mathbf{WHS}}}. \tag{39}$$

where $\mathcal{S}'$ and $\mathcal{S}$ are the reconstructed SRI and ground truth SRI, **B** is the number of bands of hyperspectral images, and **W** and **H** are the spatial dimensions of total spectral images. Low RMSE values indicate good reconstruction performance.

The third index is the spectral angle mapper (SAM), which is defined as:

$$\text{SAM} = \frac{1}{IJ}\sum_{n=1}^{IJ}\arccos\left(\frac{\mathbf{S}_{(3)}(n,:)\mathbf{S'}_{(3)}(n,:)^{\mathrm{T}}}{\|\mathbf{S}_{(3)}(n,:)\|_2\|\mathbf{S'}_{(3)}(n,:)\|_2}\right). \tag{40}$$

where $\mathbf{S'}_{(3)}(n,:)$ and $\mathbf{S}_{(3)}(n,:)$ express respectively the fibers of the reconstructed and the ground-truth SRI. SAM measures the angles between the reconstructed and the ground-truth fibers of the SRI, and a small SAM is equivalent to good performance.

The fourth index is the relative dimensionless global error in synthesis (ERGAS), which is represented as:

$$\text{ERGAS} = 100c\sqrt{\frac{1}{IJK}\sum_{k=1}^{K}\frac{\|\mathbf{S'_k} - \mathbf{S_k}\|_F^2}{\mu_k^2}}. \tag{41}$$

where $c = \frac{I}{i} = \frac{J}{j}$ is the spatial downsampling factor and $\mu_k$ is the mean values of the elements in $\mathbf{S_k}$. After image reconstruction, we hope to get a smaller ERGAS.

The fifth index is the universal image quality index (UIQI), which is defined as:

$$\text{UIQI} = \frac{1}{S}\sum_{i=1}^{S}\text{UIQI}(\mathbf{S}'^i, \mathbf{S}^i). \tag{42}$$

where:

$$\text{UIQI}(\mathbf{S}'^i, \mathbf{S}^i) = \frac{1}{P}\sum_{j=1}^{P}\frac{\sigma_{\mathbf{s}_j^i\mathbf{s}_j'^i}}{\sigma_{\mathbf{s}_j^i}\sigma_{\mathbf{s}_j'^i}}\frac{2\mu_{\mathbf{s}_j^i}\mu_{\mathbf{s}_j'^i}}{\mu_{\mathbf{s}_j^i}+\mu_{\mathbf{s}_j'^i}}\frac{2\sigma_{\mathbf{s}_j^i}\sigma_{\mathbf{s}_j'^i}}{\sigma_{\mathbf{s}_j^i}+\sigma_{\mathbf{s}_j'^i}} \tag{43}$$

$\mathbf{S}_j^i$ and $\mathbf{S}_j'^i$ show the $j$th window of the $i$th band ground truth image and reconstructed image, respectively. $P$ represents the number of window positions. $\sigma_{\mathbf{s}_j^i\mathbf{s}_j'^i}$ means the sample covariance between $\mathbf{S}_j^i$ and $\mathbf{S}_j'^i$, and $\mu_{\mathbf{s}_j^i}$ and $\sigma_{\mathbf{s}_j^i}$ denote the mean value and standard deviation of $\mathbf{S}_j^i$. The range of the index is $[-1, 1]$. The larger the value of UIQI, the better the fusion effect.

The sixth index is the normalized mean squared error (NMSE), which is represented as:

$$\text{NMSE} = \frac{\|\mathbf{S}_3' - \mathbf{S}_3\|_\text{F}}{\|\mathbf{S}_3\|_\text{F}}. \tag{44}$$

where $\mathbf{S}_3'$ and $\mathbf{S}_3$ are the three-mode unfolding matrix of the reconstructed and ground-truth SRI. The smaller the NMSE, the closer the effect of image fusion is to the ground truth image.

In addition to the above evaluation indicators, the other simplest performance indicator is the running time of the algorithm. In this paper, the efficiency of several reconstruction algorithms is compared with the computational time.

*5.3. Selection Of Parameters*

In this section, we select different iterations and the number of components of CP decomposition and experimented under these conditions to obtain the best parameter values, so as to evaluate the sensitivity of the **JTF** algorithm to important parameters in the model. Because the algorithm in this paper is based on the unknown $\mathbf{D_H}$, we first consider experimenting under the condition that the algorithm incorrectly assumes a $7 \times 7$ Gaussian blur kernel instead of using the correct $9 \times 9$ Gaussian kernel. Certainly, we have also made experimental comparisons under the correct Gaussian kernels.

Considering the **JTF** algorithm under different signal-to-noise ratios and assuming that the signal-to-noise ratio (SNR) of the HSI and MSI is the same, where the SNR here is set 20 db, to evaluate the effect of the number of iterations **Iter** on the image fusion in the algorithm, we run the **JTF** algorithm based on the number of iterations **Iter**. Because the **JTF** algorithm is based on the modification of the **Blind Stereo** algorithm [44], we compare the performance of the algorithms with or without noise of the degenerate operator. Figure 3 shows the evaluation metrics of SRI after the reconstruction of Pavia University with the change of iteration number **Iter**. In order to reduce the running time of the algorithm, without loss of generality, here $R = 100$, $\beta = 1$, where the black line represents **Blind Stereo** algorithm performance in matrix $\mathbf{D_M}$ with noise, the red is the reconstruction results of the **JTF** algorithm under the same condition, and the blue trend line indicates the fusion performance by the **Blind Stereo** algorithm when $\mathbf{D_M}$ does not contain noise.

As can be seen from Figure 3, when **Iter** changes from one to 20, the R-SNR of Pavia University decreases, while the values of NMSE, ERGAS, and SAM increase. Among them, R-SNR declines sharply and then rises, while the other evaluation metrics show the opposite trend. When the number of iterations is less than five, the reconstruction effect is better. Therefore, the maximum number of iterations of the **JTF** algorithm is set between one and five. In addition, the reconstruction effect of this algorithm is always superior to the Blind stereofusion algorithm with noise.

Then, we change the number of components from $R = 50$ to $R = 600$ to observe the effect of the number of tensor decomposition components on the image fusion, which depicts different evaluation metrics of the recovered HSIs for Pavia University; see Figure 4.

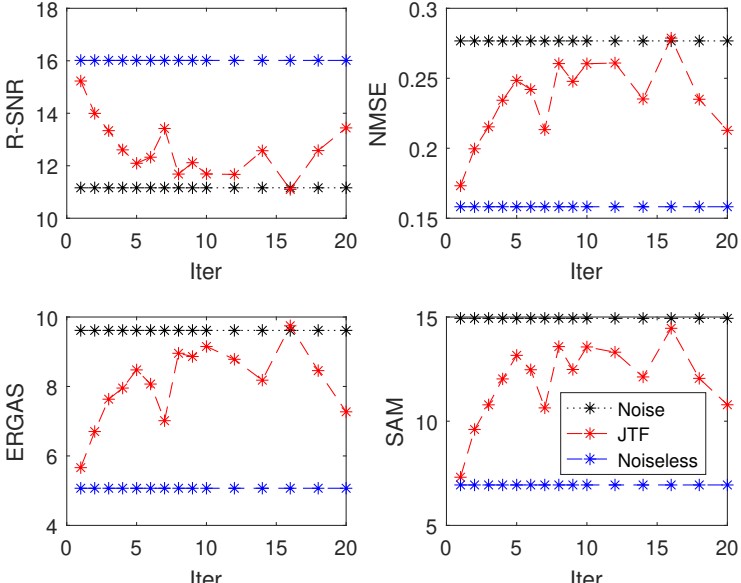

**Figure 3.** The results of the evaluation criterion as functions of the number of iterations **Iter** for the proposed joint tensor decomposition (JTF) method. SAM, spectral angle mapper; ERGAS, relative dimensionless global error in synthesis; R-SNR, reconstruction signal-to-noise ratio.

Figure 4 shows the effect of image fusion in three cases with a different number of components. When $\mathbf{D_H}$ is unknown and $\mathbf{D_M}$ contains noise, we compare the above three cases. From the six evaluation metrics, it can be seen that when the number of components is less than 100, the performance of the proposed algorithm is almost the same as the case that $\mathbf{D_M}$ is clean. It completely achieves the denoising effect and is always better than the **Blind Stereo** algorithm in the same situation. With the increase of the number of components, all three cases show good performance. However, when the number of components increases to more than 300, the reconstruction effect has a downward trend. According to the above Theorem [47], this is because when the number of components does not satisfy Theorem 1, the algorithm cannot guarantee the uniqueness of CP decomposition, which affects the initialization of the initial factor matrix in the algorithm, resulting in the poor performance of the algorithm.

Then, consider the selection range of the number of components under the condition of the uniqueness of tensor decomposition. As the **JTF** algorithm only decomposes MSI by CP, we set $I = 608$, $J = 336$, $K = 4$ in Theorem 1. According to the dimension of MSI, we can divide the selection of parameter $R$ into the following four cases:

(1) When $R < K = 4$, bring $R$ into Proposition 1, i.e.,

$$R \leq \frac{1}{2}(R + R + R - 2) \Rightarrow R \geq 1. \tag{45}$$

By synthesizing the formulas and conditions, we can get the range of R in the first case, which is $1 \leq R < 4$.

(2) When $4 = K \leq R < J = 336$, bring $R$ into Proposition 1, i.e.,

$$R \leq \frac{1}{2}(R + R + 4 - 2) \Rightarrow R \leq R + 1. \tag{46}$$

The above derivation is obviously valid, so we only consider the conditions, and we can get the range of R in the second case, which is $4 \leq R < 336$.

(3) When $J = 336 \leq R < I = 608$, bring $R$ into Proposition 1, i.e.,

$$R \leq \frac{1}{2}(R + 336 + 4R - 2) \Rightarrow R \geq 338. \tag{47}$$

By synthesizing the formulas and conditions, we can get the range of R in the third case, which is $336 \leq R < 338$.

(4) When $R \geq 608 = I$, bring $R$ into Proposition 1, i.e.,

$$R \leq \frac{1}{2}(608 + 336 + 4R - 2) \Rightarrow R \geq 475. \tag{48}$$

As such, we can get the range of $R$ in the fourth case, which is $R < 475$ and $R \geq 608$ by synthesizing the formulas and conditions. Therefore, we can conclude that there is a contradiction between the deduced range of $R$ and the range of conditions, so this situation does not exist.

To sum up, combining the above four cases, in order to guarantee the uniqueness of CP decomposition, the range of the number of components is $1 \leq R \leq 338$. In this paper, according to the fusion effect of the algorithm while ensuring the uniqueness of CP decomposition, we fixed $R = 275$.

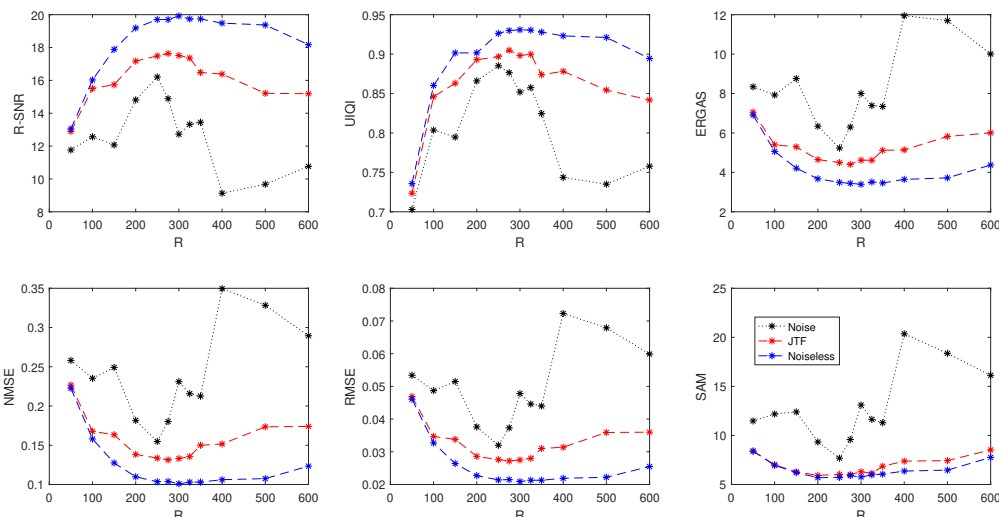

**Figure 4.** The results of evaluation metrics as functions of the number of components $R$ for the proposed JTF method. UIQI, universal image quality index.

### 5.4. Experimental Results

To further investigate the performance of the method, we conduct experiments under the incorrect Gaussian kernel ($3 \times 3$, $5 \times 5$, $7 \times 7$) and correct Gaussian kernel ($9 \times 9$) and show the fusion effect of the six test methods on Pavia University. Table 1 shows the R-SNR, NMSE, RMSE, ERGAS, SAM, and UIQI of the HSI recovered from Pavia University, and we present the best of the six algorithms in bold. As can be seen from the table, in the case of incorrect estimation of the Gaussian kernel, the fusion effect of other algorithms excluding **JTF** is worse than that of the correct Gaussian kernel. The closer these five algorithms are to the correct Gaussian kernel, the better the results will be. Nevertheless, the **JTF** method performs best in the comparison of the methods in terms of reconstruction accuracy whether the Gaussian kernel is correctly estimated or not. Overall, the **JTF** and **CNMF** methods are very effective in the reconstruction of Pavia University. On the contrary, the **JTF** algorithm proposed does not degrade the image reconstruction effect due to the incorrect estimation of the Gaussian kernel. More specifically, the property of the proposed algorithm is greater under the hypothetical Gaussian kernel, which also proves that the **JTF** algorithm has more generalization significance and application prospects.

**Table 1.** Quantitative results of the test methods on Pavia University under the different Gaussian kernels. CNMF, coupled nonnegative matrix factorization; SFIM, smoothing filter based intensity modulation; MTF-GLP, modulation transfer function based generalized Laplacian pyramid; MAPSMM, maximum a posterior estimation with a stochastic mixing model.

| Gaussian Kernel | Method | R-SNR | NMSE | RMSE | ERGAS | SAM | UIQI |
|---|---|---|---|---|---|---|---|
| $3 \times 3$ | JTF | **17.8267** | **0.1284** | **0.0266** | **4.3495** | **5.95** | **0.9031** |
| | Blind STEREO | 11.0252 | 0.281 | 0.0581 | 9.9924 | 15.841 | 0.7777 |
| | CNMF | 15.8798 | 0.1607 | 0.0332 | 5.684 | 7.8655 | 0.8922 |
| | SFIM | 10.7811 | 0.289 | 0.0598 | 10.3849 | 11.2828 | 0.7374 |
| | MTF-GLP | 12.8459 | 0.2279 | 0.0471 | 7.608 | 10.4577 | 0.7845 |
| | MAPSMM | 11.8642 | 0.2551 | 0.0528 | 8.3346 | 10.4203 | 0.7449 |
| $5 \times 5$ | JTF | **17.6357** | **0.1313** | **0.0272** | **4.4351** | **5.9266** | **0.8983** |
| | Blind STEREO | 11.163 | 0.2766 | 0.0572 | 9.6563 | 15.6988 | 0.7938 |
| | CNMF | 15.5546 | 0.1668 | 0.0345 | 5.7987 | 7.9809 | 0.8623 |
| | SFIM | 12.6416 | 0.2333 | 0.0483 | 7.9097 | 10.2705 | 0.7725 |
| | MTF-GLP | 13.6742 | 0.2072 | 0.0428 | 6.9561 | 9.7086 | 0.8038 |
| | MAPSMM | 12.8407 | 0.228 | 0.0472 | 7.4954 | 9.5283 | 0.7733 |
| $7 \times 7$ | JTF | **17.7156** | **0.1301** | **0.0269** | **4.4008** | **5.8517** | **0.9001** |
| | Blind STEREO | 13.4084 | 0.2136 | 0.0442 | 7.421 | 11.9428 | 0.8584 |
| | CNMF | 16.0028 | 0.1584 | 0.0328 | 5.5489 | 7.2096 | 0.8747 |
| | SFIM | 13.1113 | 0.221 | 0.0457 | 7.493 | 9.9524 | 0.7804 |
| | MTF-GLP | 13.9393 | 0.2009 | 0.0416 | 6.7583 | 9.461 | 0.8057 |
| | MAPSMM | 13.1616 | 0.2197 | 0.0454 | 7.2442 | 9.3344 | 0.7777 |
| $9 \times 9$ | JTF | **17.603** | **0.1318** | **0.0273** | **4.4091** | **5.7811** | **0.8991** |
| | Blind STEREO | 13.7402 | 0.2056 | 0.0425 | 7.0851 | 11.3721 | 0.8621 |
| | CNMF | 16.3576 | 0.1521 | 0.0315 | 5.3307 | 7.1092 | 0.8792 |
| | SFIM | 13.3654 | 0.2147 | 0.0444 | 7.2704 | 9.7603 | 0.7875 |
| | MTF-GLP | 14.1333 | 0.1965 | 0.0406 | 6.6077 | 9.3225 | 0.8109 |
| | MAPSMM | 13.2983 | 0.2163 | 0.0447 | 7.1304 | 9.1704 | 0.7813 |

Figure 5 reveals the fusion experimental results for Pavia University under the incorrect Gaussian kernel ($7 \times 7$), which contains the 50th and 100th bands' fused images and the corresponding error images reconstructed by the six algorithms, were Line 1 and Line 2 in Figure 5 denote the fused HSIs of the 50th band and the corresponding error HSIs of each method, respectively. Moreover, Figure 5g shows the reference HSIs, while the third and forth rows show the reconstructed images for the 100th band and corresponding error images, respectively. Except for the last column, each column in Figure 5 shows the experimental results corresponding to each method. The error image reflects the difference between the fusion result and the ground truth. As depicted in Figure 5, this paper uses the red box to show the more obvious areas in order to compare the difference of error images of different algorithms clearly. By visualized comparison of the fused HSIs with the reference HSIs, the fusion result of the **Blind STEREO** method shows slight spectral distortion on the top of the building, while the **MAPSMM** method generates fuzzy spatial details in some areas, and the spatial information of the fused image is well enhanced by the **CNMF** method. A closer inspection reveals that the spectral and spatial differences of fused HSIs obtained by the six methods are not obvious. Therefore, in order to further compare the performance of each fusion method, the second and fourth lines of Figure 5 show the error images of the six methods under two spectral bands. The error image is the difference (absolute value) between the fused HSI and the reference HSI pixel value. We magnify the data element value in the error image by 10 times, so that we can inspect it more carefully. It can be seen that the **Blind STEREO**, **SFIM**, **MTF-GLP**, and **MAPSMM** methods have large differences, while the **CNMF** method generates relatively smaller differences, and the **JTF** method has the smallest differences in most regions, indicating that this method has good fusion ability and provides clearer spatial details than the other five algorithms.

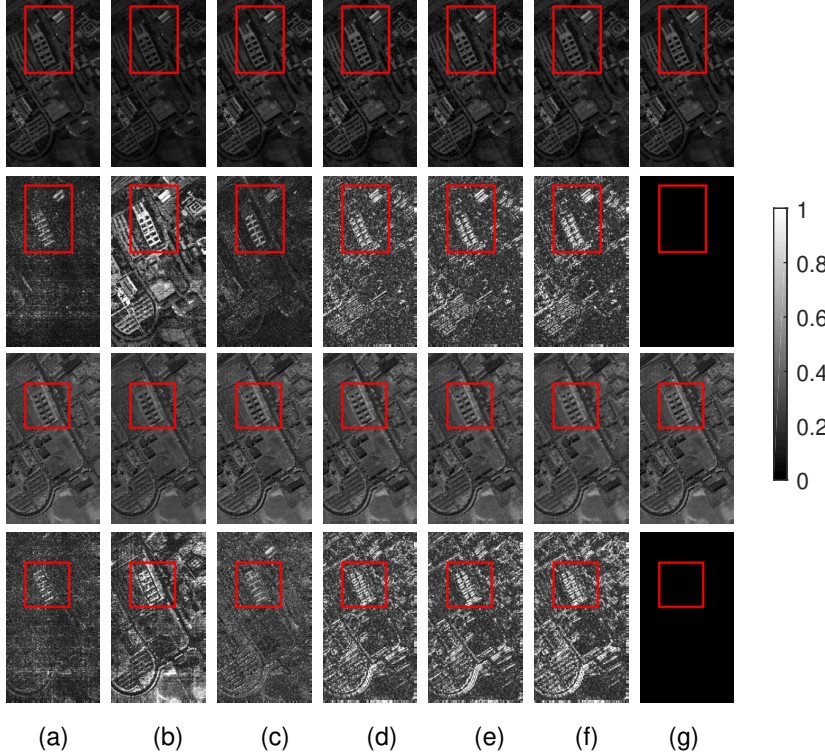

(a)     (b)     (c)     (d)     (e)     (f)     (g)

**Figure 5.** Reconstructed images and corresponding error images of Pavia University for the 50th and 100th bands with unknown $\mathbf{D_H}$ and noisy $\mathbf{D_M}$: (**a**) JTF; (**b**) Blind STEREO; (**c**) CNMF; (**d**) SFIM; (**e**) MTF-GLP; (**f**) MAPSMM; (**g**) ground truth.

Similar to the previous experiments, Figure 6 shows the fusion experimental results for Pavia University under the correct Gaussian kernel ($9 \times 9$), which contain the 50th and 100th bands' fused images and the corresponding error images reconstructed by the six algorithms. Figure 6 shows the fused HSIs of the 50th band and the corresponding error HSIs of each method, which are displayed in Lines 1–2. Moreover, Figure 6g shows the reference HSIs, while the third and forth rows show the reconstructed images for the 100th band and corresponding error images, respectively. Similarly, in order to compare the difference of the error images of different algorithms clearly, the data element values in the error image are magnified 10 times, and the red box is applied to display the region with obvious errors. The spectral distortion caused by the **Blind STEREO** method is very obvious and is affected by the Gaussian kernel changes, as shown in Figure 6b. Compared with the **Blind STEREO** method, other methods can effectively improve the spatial performance while maintaining the spectral information, and the difference between the fused images is not significant. Therefore, in order to further verify the fusion performance of the proposed method, the second and fourth lines of Figure 6 show the error images corresponding to different methods, respectively. It can be seen that the error image obtained by the **JTF** method is the lowest in most regions, and the fusion effect is not affected by the Gaussian kernel, which indicates that the **JTF** method has a superior image reconstruction effect and is more robust. Overall, the **JTF** method has better reconstruction performance and clearer fusion effects than the other five algorithms.

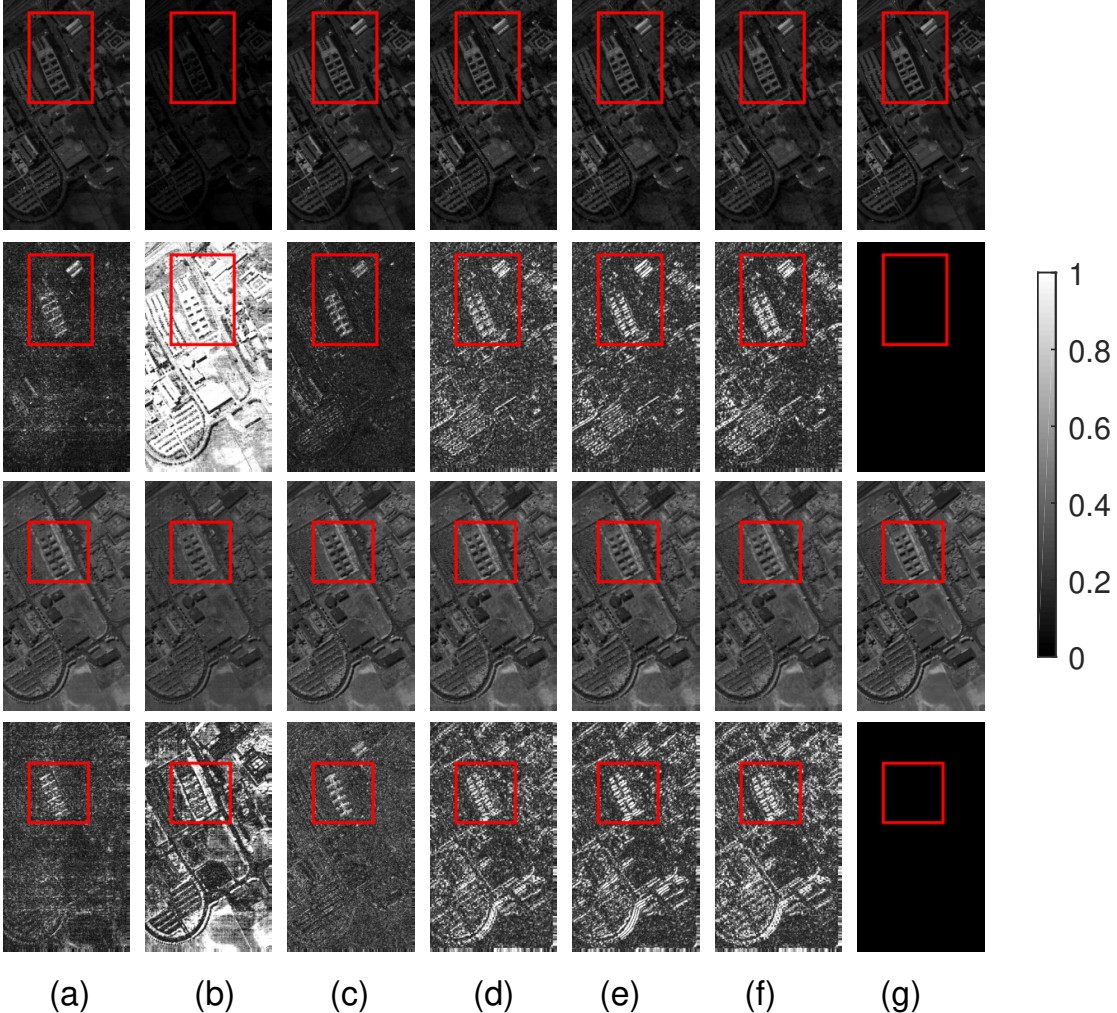

(a)  (b)  (c)  (d)  (e)  (f)  (g)

**Figure 6.** Reconstructed images and corresponding error images of Pavia University for the 20th and 60th bands with unknown $\mathbf{D_H}$ and noisy $\mathbf{D_M}$: (**a**) JTF; (**b**) Blind STEREO; (**c**) CNMF; (**d**) SFIM; (**e**) MTF-GLP; (**f**) MAPSMM; (**g**) ground truth.

### 5.5. Experimental Results of the Noisy Case

In practice, there exits additive noise in the hyperspectral and multispectral imaging processes. Therefore, to test the robustness of the proposed **JTF** method to the noise, we firstly simulate the tensor images $\mathcal{M}$ and $\mathcal{H}$ in the same way as the previous experiments for Pavia University and then add Gaussian noise to the HSI and MSI. Because the noise level in the HSI is often higher than that of the MSI, we fix the SNR added to the HSI to be 20db and compare the evaluation indicators with the traditional five classical models with the change of noise added to MSI.

Figure 7 presents the quality metric values of the noisy cases on Pavia University. It can be seen that from the reconstruction performance of the six fusion algorithms emerges a trend of enhancement with the increase of MSI image noise. Although the fusion effect of **CNMF** is closer to that of the **JTF** algorithm when the noise is high, the **JTF** method is still better than other test methods in the case of noise as a whole.

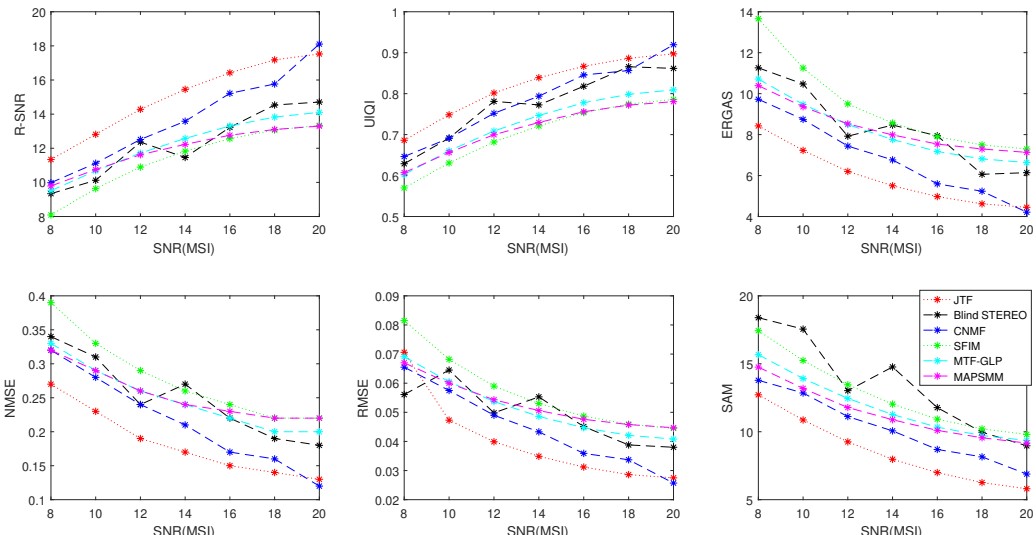

**Figure 7.** The results of evaluation metrics under different noises. MSI, multispectral image.

### 5.6. Analysis of Computational Costs

In this section, experiments are carried out on six classical methods to demonstrate the computational efficiency of the proposed method, which are accomplished with MATLAB R2016b on a PC with Intel Core i7-7500 CPU and 8 GB RAM. The mean time (in terms of seconds) of all comparison methods is shown as Table 2.

As can be seen from Table 2, the method based on the filter fusion (**SFIM**) does not need to calculate the optimal factor matrix of each mode, and its running time is shorter than the method based on tensor samples. For tensor based methods, since the iterative strategy is used to obtain the optimal solution of each unknown factor, the time of the two methods (**JTF**, **Blind STEREO**) is almost the same. **MTF-GLP** runs between the first two classes of methods, while **CNMF** and **MAPSMM** have long running times. Compared with the excellent performance, the running time of **JTF** is acceptable. Analyses were conducted based on various noises under different Gaussian kernels to further observe the performance of different algorithms. The results indicate that the running time of most algorithms is shorter under the premise of correct estimation of Gaussian kernels. However, there is little difference with the run time of the **JTF** algorithm under unknown Gaussian kernels, and the running time has little relation with the magnitude of additive noise, which indirectly proves that our algorithm is more robust.

**Table 2.** Time (s) of the test methods on Pavia University under the different conditions.

| Method | Gaussian Kernel ($7 \times 7$) SNR (10 db) | Gaussian Kernel ($9 \times 9$) SNR (10 db) | Gaussian Kernel ($7 \times 7$) SNR (20 db) | Gaussian Kernel ($9 \times 9$) SNR (20 db) |
|---|---|---|---|---|
| JTF | 17.266912 | 15.883985 | 15.303149 | 15.035694 |
| Blind STEREO | 15.179482 | 13.761605 | 13.057704 | 12.79495 |
| CNMF | 91.838425 | 89.438305 | 82.124058 | 82.466387 |
| SFIM | 1.421629 | 0.821533 | 0.836197 | 0.829191 |
| MTF-GLP | 24.859017 | 25.753572 | 24.809271 | 31.731484 |
| MAPSMM | 301.682105 | 283.545812 | 266.800409 | 299.233003 |

## 6. Conclusions

In this paper, a joint tensor decomposition method was proposed to fuse hyperspectral and multispectral images to address the hyperspectral super-resolution issue. The JTF algorithm regards the fusion problem as the joint tensor decomposition, which not only ensures the non-uniqueness of decomposition, but is applicable to the circumstance that degenerate operators are unknown or tough to gauge. In order to observe the reconstruction effect of this method, we compare the performance

of the proposed algorithm with that of the five algorithms. Experiments show that the proposed algorithm has great performance advantages and certain simulation prospects. For our future work, we would concentrate on the novel scenario that adds the non-negative constraints for the joint tensor decomposition of super-resolution images.

**Author Contributions:** Methodology, X.R. and L.L.; resources, J.C.; validation, X.R.; writing, original draft, X.R. and L.L.; writing, review and editing, L.L. All authors read and agreed to the published version of the manuscript.

**Funding:** The authors would like to thank the Editors and anonymous Reviewers. This work was partially supported by the National Natural Science Foundation of China under No.51877144.

**Acknowledgments:** The authors would like to thank the Editors and Reviewers of the Remote Sensing journal for their constructive comments and suggestions.

**Conflicts of Interest:** The authors declare no conflict of interest.

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
