# Peer review of "Toward Super-Resolution Image Construction Based on Joint Tensor Decomposition"

_remotesensing, doi:10.3390/rs12162535_

Round 1

Reviewer 1 Report

The paper entitled with “Toward Super-Resolution Images Construction Based on Joint Tensor Decomposition” proposed to construct super-resolution images based on joint tensor decomposition. I have the following comments to be addressed.

--Line 26: it is not correct to say “it is often arduous to earn high-resolution hyperspectral images (HSR) due to the hardware constraints”, how about hyerspectral images captured UAVs?

--Line 33: the three categories seems arbitrary, what is the basis to divide these three categories?  In addition, “the hybrid method” is one category, however the name provides nothing to readers what it is.

--Line 38: it is also to absolute to say “ the MSI has higher spatial resolution than HSI”

--Line 49: “Compared with HSI and Brovey transform, “ what does this sentence mean?

--Lines 65, 74 and others: please avoid to use authors’ full name to reference on papers, such as “Naoto Yokoya “, “Charis Lanaras”

--Line 79: please re-organize the introduction part: the authors have already introduced several advancements, like [25]-[29], and then introduced some very classic work, [30] and [32].

--Line 101: please provide some introduction on the “degenerate operators”, it is very hard for readers to follow this sentence.

--Line 119: “six test methods at the Pavia University as well” , please to be more specific to refer to the dataset, not the university.

--Lines 179-192: please clearly define the data you used: we know the Pavia dataset is 610×310×103, the original Pavia data as SRI, and Gaussian blurred one as HSI?

--Figs.5-6:  from the figures, it’s hard to see differences among difference algorithms, please only provide the figure for the subset of data the authors actually used.

-- Please add another group of experiments, only one data set is not convincingly demonstrate the effectiveness of the proposed algorithm.

Author Response

Responses to Reviewer 1

General Comments: The paper entitled with “Toward Super-Resolution Images Construction Based on Joint Tensor Decomposition” proposed to construct super-resolution images based on joint tensor decomposition. I have the following comments to be addressed.

Response: Thank you very much for your encouraging words.

Comment-1-1: --Line 26: it is not correct to say “it is often arduous to earn high-resolution hyperspectral images (HSR) due to the hardware constraints”, how about hyerspectral images captured UAVs?

Response-1-1: Thank you very much for your suggestion. It’s our mistake for the inappropriate presentation. The revised content is listed as follows. (Please see line 26 on page 1)

“However, the spatial resolution of HSI is still relatively low subjected to the imaging equipment of HSI and the complex imaging environment, which cannot meet the application requirements of mixing, classification, detection, etc., while it further limits the prospect of HSI.”

Comment-1-2: --Line 33: the three categories seems arbitrary, what is the basis to divide these three categories?  In addition, “the hybrid method” is one category, however the name provides nothing to readers what it is.

Response-1-2: We appreciate the suggestions. We mainly refer the reference “Multi-resolution analysis techniques and nonlinear PCA for hybrid pansharpening applications” proposed by Licciardi, G in Multidimensional Systems and Signal Processing Journal. They classified the pansharpening techniques into component substitution (CS) and multi-resolution analysis (MRA). Meanwhile, they proposed a hybrid method, combining the better spatial information of CS and the more accurate spectral information of MRA techniques, to improve the spatial resolution while preserving as much as possible of the original spectral information. The relevant contents are shown as follows.  (Please see line 34 on page 2)

“In [14], the authors classified the pansharpening techniques into component substitution (CS) [15] and multi-resolution analysis (MRA) [16]. Meanwhile, they proposed a hybrid method, combining the better spatial information of CS and the more accurate spectral information of MRA techniques, to improve the spatial resolution while preserving as much as possible of the original spectral information.

  1. Licciardi, G.; Vivone, G.; Dalla Mura, M.; Restaino, R.; Chanussot, J. Multi-resolution analysis techniques and nonlinear PCA for hybrid pansharpening applications. Multidimensional Systems and Signal Processing 2015, 27.
  2. Shah V P, Y.N.H.; L, K.R. An efficient pan-sharpening method via a combined adaptive PCA approach and contourlets. IEEE Transactions on Geoscience and Remote Sensing 2008, 46, 1323–1335.
  3. P, N.G.; W, S.B. The stationary wavelet transform and some statistical applications. IEEE Transactions on Geoscience & Remote Sensing 1995, 346, 918–9.”

Comment-1-3: --Line 38: it is also to absolute to say “ the MSI has higher spatial resolution than HSI”

Response-1-3: We are grateful for your careful review. We have modified the relevant sentence. The revised content is listed as follows. (Please see line 42 on page 2)

“Generally, the MSI has higher spatial resolution than HSI, which is complementary to the HSI.”

Comment-1-4: --Line 49: “Compared with HSI and Brovey transform, “ what does this sentence mean?

Response-1-4: Thank you very much for your suggestion. We have modified the relevant contents, which are shown as follows. (Please see line 52 on page 2)

“Compared with Brovey transform [23], SFIM is an advanced fusion technology to improve the spatial details of MSI, and its spectral characteristics are reliably preserved.

  1. Zhijun Wang.; Ziou, D.; Armenakis, C.; Li, D.; Qingquan Li. A comparative analysis of image fusion methods. IEEE Transactions on Geoscience and Remote Sensing 2005, 43, 1391–1402.”

Comment-1-5: --Lines 65, 74 and others: please avoid to use authors’ full name to reference on papers, such as “Naoto Yokoya “, “Charis Lanaras”

Response-1-5: Thank you very much for your suggestion. We have modified this mistake. (Please see line 72 on page 2)

“In the effort [29], the authors proposed the method of enhancing spatial resolution of HSI in terms of unmixing technology: coupled nonnegative matrix factorization (CNMF).

To further boost the effect of super-resolution reconstruction, the authors of [31] came up with a method to resolve the problem of super-resolution and hyperspectral unmixing simultaneously.

  1. Yokoya N, M.S. Coupled nonnegative matrix factorization unmixing for hyperspectral and multispectral data fusion. IEEE Transactions on Geoscience and Remote Sensing 2012, 50, 528–537.
  2. Lanaras C, B.E.; K, S. Hyperspectral super-resolution by coupled spectral unmixing. IEEE International Conference on Computer Vision, 2015, pp. 3586–3594.”

Comment-1-6: --Line 79: please re-organize the introduction part: the authors have already introduced several advancements, like [25]-[29], and then introduced some very classic work, [30] and [32].

Response-1-6: Thank you very much for your suggestion. We have rearranged the structure of the introduction part. (Please see line 59 on page 2)

Comment-1-7: --Line 101: please provide some introduction on the “degenerate operators”, it is very hard for readers to follow this sentence.

Response-1-7: We are grateful for your careful review. The degenerate operators represent the matrices applied to SRI to form corresponding HSI and MSI.

Comment-1-8: --Line 119: “at the Pavia University as well”, please to be more specific to refer to the dataset, not the university.

Response-1-8: Thank you very much for your suggestion. We have modified the relevant contents, which are shown as follows. (Please see line 119 on page 3)

“Besides, we conduct experiments under the incorrect gaussian kernel (3× 3, 5× 5, 7× 7), correct gaussian kernel (9× 9) and different noises, while show the fusion effect of the six test methods with the Pavia University data captured by the ROSIS sensor as well.”

Comment-1-9: --Lines 179-192: please clearly define the data you used: we know the Pavia dataset is 610×310×103, the original Pavia data as SRI, and Gaussian blurred one as HSI?

Response-1-9: Thanks for your suggestions. The revised contents are shown as follows. (Please see line 188 on page 10)

“The data selected in this paper was taken from the Pavia University in Italy and captured by the ROSIS sensor. The SRI, HSI and MSI are with sizes of 608×336×103, 152×84×103 and 608×336×4, respectively. Specifically, MSI is generated by Quickbird simulation, while HSI is generated by SRI by 9×9 gaussian blur and downsampling, and MSI is generated for the Pavia University image according to the QuickBird specification. The degradation process from SRI to HSI is a combination of spatial blurring of 9×9 gaussian kernel and D=4 factor along two spatial directions to model the blurred image.”

Comment-1-10: --Figs.5-6:  from the figures, it’s hard to see differences among difference algorithms, please only provide the figure for the subset of data the authors actually used.

Response-1-10: We are grateful for your comment. Figure 5 and Figure 6 reveals the fusion experimental results for the Pavia University. In fact, these restructured images are different. In order to compare the difference of error images of different algorithms clearly, we use the red box to show the more obvious areas. Moreover, Table 1 shows the relevant metrics of HSI recovered by Pavia University in numerical form.

Comment-1-11: -- Please add another group of experiments, only one data set is not convincingly demonstrate the effectiveness of the proposed algorithm.

Response-1-11: Thanks for your suggestions. We understand that more experimental data may better reveal the robustness of the proposed method. However, in the present study, we mainly focused on circumstance that degenerate operators are unknown or tough to gauge, and we consider that the existing experiments may not be optimal, but should be sufficient to draw a conclusion that the proposed algorithm has great performance advantages and certain simulation prospects under mild and realistic conditions. The extended experiments with related methods under investigation in our laboratory. Unfortunately, results are unavailable at this point. And we would enhance the robustness of the algorithm in the following research work.

We would express our thanks again to both anonymous reviewers for your thoughtful and thorough review and all the editors for your detailed and prompt reply. We hope this revised manuscript will be convincing under your constructive guidelines and we’re looking forward to your positive reply.

Best Regards,

Authors

Reviewer 2 Report

The authors added some elements to enrich manuscript and I think this paper can now be accepted for publication.

Author Response

Responses to Reviewer 2

General Comments - The authors added some elements to enrich manuscript and I think this paper can now be accepted for publication.

Response: Thank you very much for your encouraging words. We are very grateful for the precious time and effort you have spent on our paper. Meanwhile, we are honored to receive your recognition and support for our work. Moreover, we have corrected some typos and grammatical errors. The relevant modified content is shown as follows.

“The matrix is denoted as X, and the scalar (or the vector) is represented by x.

One result of the Kruskar criterions is the following statement, which applies to general tensors, which provides the uniqueness proof of the CP decomposition model.

We use the following optimization models to obtain factor matrices A,B and C. Where  is the regularization parameter.

Where line 1 and line 2 in Figure 5 denote the fused HSIs of the 50th band, and the corresponding error HSIs of each method, respectively.

In practice, there exits additive noise in the hyperspectral and multispectral imaging processes.”

We would express our thanks again to both anonymous reviewers for your thoughtful and thorough review and all the editors for your detailed and prompt reply. We hope this revised manuscript will be convincing under your constructive guidelines and we’re looking forward to your positive reply.

Best Regards,

Authors

Reviewer 3 Report

The Authors have responded correctly to almost all of my remarks and suggestions.
The quality of the content of this manuscript has been improved and is now acceptable for publication.
I noticed some small residual details/typos to be corrected while preparing the final version (for publication)

please add a reference following "... and Brovey transform,"

please substitute " the optimization with respect to (w.r.t.) C in (22) " for "the optimization with respect to (w.r.t.) A in (22)"

please add a blank when necessary: Stereo[43] ... CNMF(C ... Factorization)[25] ... SFIM(S ... MTF-GLP(M ... MAPSMM(M ...

Typo(s):
... (or. vector) ...
... Kruskar criterion ...
... A, B, C. Where β is regularization parameter. ...
... Figure 5 (line 1-2) show ...
... in the hyperspectral and multispectral imaging process ...

Author Response

Responses to Reviewer 3

General Comments - The Authors have responded correctly to almost all of my remarks and suggestions. The quality of the content of this manuscript has been improved and is now acceptable for publication. I noticed some small residual details/typos to be corrected while preparing the final version (for publication)

Response: Thank you very much for your encouraging words.

Comment-3-1: please add a reference following "... and Brovey transform,"

Response-3-1: Thank you very much for your suggestion. We have added the relevant presentation. The relevant contents are as follows: (Please see Line 52 on page 2)

“Compared with HSI and Brovey transform [23], SFIM is an advanced fusion technology to improve the spatial details of MSI, and its spectral characteristics are reliably preserved.

  1. Zhijun Wang.; Ziou, D.; Armenakis, C.; Li, D.; Qingquan Li. A comparative analysis of image fusion methods. IEEE Transactions on Geoscience and Remote Sensing 2005, 43, 1391–1402.”

Comment-3-2: please substitute " the optimization with respect to (w.r.t.) C in (22) " for "the optimization with respect to (w.r.t.) A in (22)"

Response-3-2: Thanks for your suggestions. We have modified this mistake. The relevant descriptions are marked by red color in Section 4.1. (Please see page 8)

Comment-3-3: please add a blank when necessary: Stereo[43] ... CNMF(C ... Factorization)[25] ... SFIM(S ... MTF-GLP(M ... MAPSMM(M ...

Response-3-3: We are grateful for your comment. We modified the relevant format. (Please see Line 204 on page 10)

“To further demonstrate the performance of our proposed algorithm, this method is compared with the following five HSI-MSI fusion methods: Blind Stereo [44], CNMF (Coupled Nonnegative Matrix Factorization) [29], SFIM (Smoothing Filter-based Intensity Modulation) [51], MTF-GLP (Modulation Transfer Function based Generalized Laplacian Pyramid) [19] and MAPSMM (Maximum a Posterior Estimation with a Stochastic Mixing Model) [24].

  1. Aiazzi B, A.L.; S, B. MTF-tailored multiscale fusion of high-resolution MS and pan imagery. Photogrammetric Engineering and Remote Sensing 2006, 72, 591–596.
  2. T, E.M. Resolution enhancement of hyperspectral imagery using maximum a posteriori estimation with a stochastic mixing model. Dissertation Abstracts International 2004, 65, 1385.
  3. Yokoya N, M.S. Coupled nonnegative matrix factorization unmixing for hyperspectral and multispectral data fusion. IEEE Transactions on Geoscience and Remote Sensing 2012, 50, 528–537.
  4. Kanatsoulis C I, F.X.; D, S.N. Hyperspectral super-resolution: a coupled tensor factorization approach. IEEE Transactions on Signal Processing 2018, 66, 6503–6517.
  5. G, L.J. Smoothing filter-based intensity modulation: a spectral preserve image fusion technique for improving spatial details. International Journal of Remote Sensing 2000, 21, 3461–3472.”

Comment-3-4: Typo(s):

... (or. vector) ...

... Kruskar criterion ...

... A, B, C. Where β is regularization parameter. ...

... Figure 5 (line 1-2) show ...

... in the hyperspectral and multispectral imaging process ...

Comment-3-4: Thank you very much for your suggestion. We have corrected the typos and grammatical errors. In addition, we have proofread the whole manuscript carefully, and trying hard to avoid typos and grammar errors. Meanwhile, we have highlighted the major changes in red color in the revised manuscript. The relevant modified content is shown as follows.

“The matrix is denoted as X, and the scalar (or the vector) is represented by x.

One result of the Kruskar criterions is the following statement, which applies to general tensors, which provides the uniqueness proof of the CP decomposition model.

We use the following optimization models to obtain factor matrices A,B and C. Where  is the regularization parameter.

Where line 1 and line 2 in Figure 5 denote the fused HSIs of the 50th band, and the corresponding error HSIs of each method, respectively.

In practice, there exits additive noise in the hyperspectral and multispectral imaging processes.”

We would express our thanks again to both anonymous reviewers for your thoughtful and thorough review and all the editors for your detailed and prompt reply. We hope this revised manuscript will be convincing under your constructive guidelines and we’re looking forward to your positive reply.

Best Regards,

Authors

Round 2

Reviewer 1 Report

No other comments.

This manuscript is a resubmission of an earlier submission. The following is a list of the peer review reports and author responses from that submission.

Round 1

Reviewer 1 Report

In this manuscript, the Authors propose a modification of the Blind Stereo algorithm for solving the hyperspectral super-resolution problem.
The method is considered as a common tensor decomposition method with the interesting property of taking into account the uncertainty on the involved (degenerate) operator used as well as additional noise.
Compared to five other methods representative from the literature, the proposed method is proved experimentally to provide an improvement, in particular against the CNMF method (second best competitor) when the Gaussian kernel operator used is not accurately known (with undetermined extent).

Overall, the manuscript is quite well written.
The Authors have provided a quite well-structured exposition of their material.
The content is novel and original. It is described with a certain amount of details to understand the topic, results and techniques.
The analysis provided and corresponding figures are quite appropriate to the text and its content. But the argument and analysis could be improved in several places.
The list of references to the literature related to this field is also quite appropriate.

I have reported some points below that can help improving further its content.

1) The writing style of a certain number of sentences needs to be reviewed and improved in order to improve the manuscript readability, which is sometimes abrupt. There are also many terms that are misused and need to be corrected.
On the other hand, the results are good and convincing.

2) However, the absence of recommendations for the adjustment of the regulation parameter beta lacks. The unstable nature of the curves (fig. 3 and fig. 4) in the noisy case suggests that the weight of the last term in (22) is not adjusted finely enough. It would be interesting to appreciate what a different setting of the parameter beta can bring in terms of impact on the quality of the final results obtained by the proposed method (JTF).

3) Note that the choice of an appropriate value of R remains also an open question in practice. The range of the number of components to guarentee the uniqueness of the CP decomposition is quite large. Could you provide a few clues to justify your particular choice (1<R=275<338)?

4) define ' in (9)

5) define what you mean exactly by "from the effect point of view"

6) It is necessary to position the pseudo-algorithm 1 correctly in relation to the text (below (37)).

7) Refine the following phrase: "while the number of iteration updates for factor matrix needs to continue numerical simulation."

8) Clarify what you mean by "The optimal number of iterations is selected according to the reconstruction effect", and justify your specific choice to fix beta=1

9) Add a reference to "According to the above theorem"

10) Rephrase "According to the fusion effect of the algorithm and ensure the uniqueness of CP decomposition, that is, the selection of the number of components should satisfy the condition of Proposition 1. we fixed R = 275 in this paper"

11) In figure 3, the behaviour of the proposed method (JTF) is not very stable wrt to the iteration number (Iter), although a smoother general trend is emerging. What are the reasons for this? I'm wondering about the setting chosen for the parameter beta.

12) In figure 4, what are the reasons for such a rowdy behavior in the noisy case?

13) Currently, the situation being tested experimentally in section 5.4 is that of an under-determination of the Gaussian kernel extent. What would happen in the case of an overdetermination of this value?

14) Rephrase "red matrix"

15) Rephrase "we enlarge the value of data elements in error images by 10 times to inspect more carefully"

16) Please, put a blank between the last character of the word before a parenthesis and the subsequent parenthesis (

17) There are a lot of typos to correct. The list below is not exhaustive. Please check carefully.
... we first introduces ...
... and the calculation results is a matrix ...
... model.At the time ...
... ,the CP decomposition ...
... ,c_r ...
... HSI(Tensor... and MSI(Tensor ...
In this section, We ...
... C. Where β is regularization parameters.
... with respect to (w.r.t.) A in (22) -> w.r.t. C (23)
When A, B, A , B and C fixed, ...
When A, B, C, B and C fixed, ...
When A, B, C, A and B fixed, ...
When B, C, A , B and C fixed, ...
... ), and red ...
... Gauss noise ...
... Gauss kernel ...
... . Where the CP decomposition involved is computed by TensorLab [51]. ...
... methods:Blind Stereo ...
... the performance of the proposed algorithm has almost the same as the DM is clean,
Where Figure 6 show the reconstructed ...
... and comparing evaluation indicators with the traditional five classical models with the change of noise added to MSI ...
For tense-based methods ...
... the time of the two methods ...
... the un-uniqueness of decomposition, ...

Reviewer 2 Report

Overall, this paper is in good shape. I agree that the new knowledge provided helps to acquire super-resolution image.
I list here below some minor comments to improve the presentation, but in general I would say that the manuscript requires minor revisions only. It would be great to see your publication in this journal.

5.1. Experimental data
Although you showed some reconstructed images, you should have shown some thumbnails of the original Quickbird imagery over your study area.

L. 220
Could you clarify the reason why you fixed R = 275?

Reviewer 3 Report

Before further review, I think the authors should first re-write their papers. When I checked the novelty of this study, I found one published paper (paper title: Exploring coupled images fusion based on joint tensor decomposition) by the same authors. However, there are some similar paragraphs between these two papers (especailly the two paragraphs in the submitted paper with respect to "Tensor and related notation" section in the already published one). In this regard, I think this paper should be rejected. 

The link for the paper "Exploring coupled images fusion based on joint tensor decomposition" 

https://hcis-journal.springeropen.com/articles/10.1186/s13673-020-00215-z

Reviewer 4 Report

This paper presents a hyperspectral image fusion method based on joint tensor decomposition algorithm. The experiment shows effectiveness. However, to improve the manuscript, some questions should be clarified.

  1. Page 3, line 94 Xiaofu should be Xiao Fu; line 96 Renwei Deng should be Renwei Dian.

  1. In this manuscript, this paper chose one tensor-based method and several classical fusion methods for comparison. Indeed, the experiment presents the advantage of the proposed method. However, more tensor decomposition and deep learning based fusion methods have also been developed in recent years. At least, the author should increase one or more tensor-based comparison methods to demonstrate the advantage of the proposed method.

  1. Only Pavia University data was used as experimental data. More experimental data are needed to verify the robustness of the proposed method.